# Habitat Diversity, Environmental Conditions, and Distribution of Endangered Fungus *Sarcosoma globosum* (Ascomycota) in Lithuania

**DOI:** 10.3390/jof10040263

**Published:** 2024-03-30

**Authors:** Eglė Vabuolė, Sigitas Juzėnas, Ernestas Kutorga

**Affiliations:** Department of Botany and Genetics, Institute of Biosciences, Life Sciences Center, Vilnius University, Saulėtekio Ave. 7, LT-10257 Vilnius, Lithuania; sigitas.juzenas@gf.vu.lt (S.J.); ernestas.kutorga@gf.vu.lt (E.K.)

**Keywords:** *Sarcosoma globosum*, threatened species, distribution range, forest management, habitat disturbance, fungal conservation, *Picea abies*

## Abstract

*Sarcosoma globosum* (Pezizales, Ascomycota) is a rare and endangered fungus, and it is believed to be extinct in most central European countries. Known records of *S. globosum* in Lithuania reveal that it is situated on the south-western edge of a shrinking geographical distribution range in Europe. An assessment of the species’ current habitat conditions and threats could enhance and provide new knowledge and guidelines to facilitate the efficient conservation of this threatened fungus and its habitats. The main aim of this study was to analyse the habitats and environmental conditions of *S. globosum* in Lithuania. We examined the diversity of habitats, various soil and tree stand characteristics, forest management activities, and natural disturbances in all 28 known fungus localities. *S. globosum* habitats in Lithuania are restricted to coniferous forests with the presence of *Picea abies*; the species was observed in boreo-nemoral bilberry western spruce taiga (the European Nature Information System habitat type T3F14), continental tall-herb western spruce taiga (T3F44), and native fir, spruce, larch, and cedar plantations (T3N1). An analysis of forest stand age structures in Lithuanian *S. globosum* localities revealed a rather large proportion of young Norway spruce stands of cultural origin (25.6% of study plots were assigned to age classes from 21 to 50 years); nevertheless, the majority of fungus growth sites were situated in older forests. Various natural and anthropogenic disturbances that threaten *S. globosum* habitats were assessed.

## 1. Introduction

There is a consensus that global climate change and environmental management practices affect the development of fungal reproductive structures, biodiversity, and species distribution [1,2,3,4]. Studies provide evidence that fungal distribution, biology, and ecology depend on a wide range of environmental factors and that fungi may be highly vulnerable to the strong impacts of land cover change, overharvesting, low moisture, and high temperatures [5,6,7,8]. Therefore, when significant declines were reported, this led to global challenges associated with fungi, including the impacts of climate change on fungi and global efforts to conserve them [9]. To date, of the 487 globally threatened fungal species for which the International Union for Conservation of Nature’s Red List tracks population trend data, 306 are recorded as decreasing (63%), 177 as stable (36%), and just 4 (1%) as increasing [10]. Thousands more fungal species should be red-listed, and their threat status and extinction risk should be assessed in order to support the identification of habitats and critical areas for fungal conservation [11,12,13,14,15].

Witches’ cauldron fungus (*Sarcosoma globosum*) is a saprotrophic ascomycete, which produces rather large, globose, 3–12 cm in diameter, dark-coloured, gelatinous apothecia (Figure 1) [16,17]. The etymology of the name derives from the Greek words *sarco* (flesh/fleshy) and *soma* (body) as well as the Latin *globosus* (round like a ball, globose, or spherical) [18]. *S. globosum* is a type of species from the genus *Sarcosoma*, the core of the Sarcosomataceae family (Pezizales, Ascomycota) [19]. The fungus has a low culinary value; however, in places of mass fruiting, it has been consumed [20] or used as a folk medicine, particularly in the Kirov region and West Siberia in Russia [21]. *S. globosum* develops its fruiting bodies on the ground in moss or decaying needle litter, mainly in habitats with spruce in the spring and early summer [22]. The fungus is considered to be an indicator species of the ecological integrity of forests and is generally associated with primeval and old-growth spruce or fir–spruce forests little affected by human activity [23,24,25].

This species is found in eastern parts of North America, Europe, and the Asian taiga zone [26]. It has been reported in several European countries, although it is not common anywhere. The decrease in its frequency and extinction occurred in the 20th century in central European countries [22]. The fungus is believed to be extinct in Austria (Mykologische Datenbank (https://www.pilzdaten-austria.eu, accessed on 24 March 2024)), Switzerland (Swiss Fungi (https://swissfungi.wsl.ch/en/, accessed on 24 March 2024)), Czechia, Slovakia, Germany, and Poland [20,27,28,29]. Since the beginning of the 20th century, *S. globosum* has not been observed in the Kaliningrad vicinities of Russia (formerly Königsberg of East Prussia) [30] (Alexandra Volodina, personal communication). In order to recognise the need for the conservation of this fungus and its habitats, the species was included in a first preliminary European Red List of endangered macrofungi [31] and was one of the thirty-three threatened species that were proposed for inclusion in the appendices of the Bern Convention [32,33]. The fungus is discussed and evaluated in the International Union for Conservation of Nature (IUCN) Global Fungal Red List as near-threatened (NT) and is nationally red-listed in 10 European countries [26]. Past, ongoing, and future habitat declines are estimated to impact negatively on all populations of *S. globosum* [26]. In contrast, an increase in sites has been reported for the last decades in Finland, Russia, and the Baltic states [34,35,36,37].

The first record of *S. globosum* in Lithuania was in 1950—the collected specimen, which is preserved in the Vilnius University Herbarium (WI), contains fungus fruiting bodies that are sold in the spring at Šiauliai market as a folk remedy called “the fat of the earth” [38]. Thereafter, the fungus has been reported in the Anykščiai, Panevėžys, Šiauliai, and Vilnius districts, noticing its use in folk medicine to treat rheumatism [39,40]. However, no records of this species were provided for a relatively long time (more than four decades) until 2007, when it was repeatedly discovered in the country [36,41]. In total, 28 species localities were documented from 2007 to 2021, mainly in the north-eastern part of Lithuania (Figure 2; Appendix A). Currently, the known localities of this fungus in Lithuania are situated on the south-western edge of a shrinking geographical distribution range for this species in Europe. The ascomycete *S. globosum* has been protected in Lithuania since 1992; it was included in the list of strictly protected animal, plant, and fungal species in the Lithuanian Republic in 2010 and evaluated as an endangered species (EN) according the IUCN criteria in 2021 [36,42]. According to the applicable national legislation, it is mandatory for protected species to have a species protection plan and, optionally, a species action plan for species protection in a specific habitat/area. However, due to various reasons, these documents have not yet been prepared. The foremost reason could be a general lack of understanding of fungus biology and ecology, threats, and possible preservation measures to mitigate further population loss. Moreover, the habitats where *S. globosum* develops are involved in decreases or changes due to climatic change, natural and anthropogenic disturbances, and a lack of proper maintenance. In that sense, finding ecological characteristics related to the presence of *S. globosum* becomes essential to preserve these environments and their ecological value. The assessment of species’ current habitat conditions and threat factors could enhance and provide new knowledge and guidelines to facilitate the efficient conservation of this threatened fungus and its habitats in situ [15,43,44,45,46,47]. New or increased knowledge can also result in different extinction risk assessments, which remain among the most powerful tools in fungal conservation science because they support and inform conservation policies, planning, and action [12]. The main aim of this study was to describe and analyse the habitat diversity and environmental conditions of *S. globosum* in Lithuania. The specific objectives were to (1) compare the characteristics of the habitats in different fungus localities, (2) investigate the distribution pattern of the fungus in Lithuanian forest ecosystems, (3) assess the natural and anthropogenic disturbances in the localities of *S. globosum*, and (4) discuss the protection of this endangered species.

## 2. Materials and Methods

### 2.1. Study Area and Design

The field studies conducted in 2021–2022 covered the eastern and north-eastern regions of Lithuania and surveyed all known *Sarcosoma globosum* localities within the country (Figure 2). We employed the European Environment Agency (EEA) reference grid set at a resolution of 10 km × 10 km to depict the distribution of *S. globosum* in Lithuania, along with study plot counts per grid cell. Additionally, detailed forest-cover data for Lithuania were incorporated to discern distribution patterns between broad-leaf and coniferous forests. Specifically, we obtained the Dominant Leaf Type 2015 raster dataset with a resolution of 20 m from the Copernicus Land Monitoring Service website (https://land.copernicus.eu/en/products/high-resolution-layer-dominant-leaf-type/dominant-leaf-type-2015, accessed on 4 April 2023). The altitude of fungus localities ranged from 61 m to 195 m above sea level. The mean annual air temperature in the study area was about 5.8 °C (the hottest month was July at 18.3 °C and the coldest month was January at −2.9 °C), the mean annual precipitation was about 675 mm, and the snow-cover duration was about 67 days a year [48,49]. The study area was in the periphery of the boreal region [50]; however, according to other authors, the study area was situated at the northern edge of the nemoral environmental zone [51]. Forests cover about 33.7% of Lithuanian territory, with Scots pine (*Pinus sylvestris*) (32.2%) and Norway spruce (*Picea abies*) (19.8%) dominating (data are from the State Forest Service database (https://amvmt.lrv.lt/lt/atviri-duomenys-1/misku-statistikos-leidiniai/valstybine-misku-apskaita/2022-01-01/, accessed on 10 July 2023)).

We established sampling plots within all 28 currently known localities of *S. globosum*, which were discovered over a period of 15 years from 2007 to 2021. Distribution data of the fungus were originally obtained from different sources. The sources of observational data included reports from environmental authorities (professional employees of protected territories and foresters) and different citizen observations. Every parent observation report was verified by us and the facts of *S. globosum* fructification in all localities were approved during field and laboratory observations. The apothecia of *S. globosum* are highly distinctive and the species does not have any known lookalikes [37]. The identification of *S. globosum* was based on macro- and micromorphological characteristics, including globose apothecia, 3–12 cm in diameter, dark-brown, thick-fleshed, and very gelatinous; an outer surface felted with short, dark-brown hyphal outgrowths; a hymenium forming a saucer-shaped depression in the upper part of the apothecium; cylindrical asci, thick-walled, 8-spored, and 250–500 × 12–15 μm; ellipsoid ascospores, smooth, with small guttules, hyaline, and 20–30 × 7–10 μm; narrow paraphyses with tips enlarged up 6 μm and brown in colour [41,52]. *S. globosum* is a protected species in Lithuania; therefore, the reference collection of fungal specimens was made in accordance with legal requirements and deposited in the mycological collection of the Vilnius University Herbarium (WI). In total, 43 sampling plots (circular, 14 m radius, and 616 m^2^ in size) were established within each forest compartment (a forest management unit with a uniform forest type) with the presence of *S. globosum*. General and environmental data of the study plots are provided in Appendix A. 

### 2.2. Characterisation of Habitats

For the analysis of chemical soil parameters, topsoil was sampled in all *S. globosum* localities in Lithuania. To minimise disturbance, the soil was taken only from one study plot per locality. The sampling of soil was performed at five points (one central point and four points on the edge of the circular study plot on the north, east, south, and west sides corresponding with azimuth angles of 0°, 90°, 180°, and 270°). After removing plant debris from the soil surface, five subsamples of soil core were taken from a uniform depth (10 cm) in the plot using a polyvinyl chloride (PVC) tube (5 cm in diameter). Five subsamples (each ~100 g) were pooled into one composite sample (~500 g). In total, 28 soil samples were air-dried at room temperature. Soil pH values, the content of organic carbon (C), and total nitrogen (N) were measured in the Agrochemical Research Laboratory of the Institute of Agriculture, The Lithuanian Research Centre for Agriculture and Forestry, following standard methods [53,54,55]. The thickness of the soil O horizon was measured using a ruler in holes dug 20–25 cm deep in all 43 study plots. The data on soil groups according to the classification of the Food and Agriculture Organization of the United Nations were extracted from the soil maps of the forest area of Lithuania [56,57]. Data of the soil characteristics are presented in Appendix A. 

We studied different environmental variables to characterise *S. globosum* habitats for each sampling plot. Field sampling was undertaken from June until July 2022. The study plots were surveyed thoroughly during one plot visit, with necessary precautions for the growth sites of the protected species. We identified the habitat types following the European Nature Information System (EUNIS) habitat classification [58]. Vertical vegetation structures were examined by assessing the cover of the canopy and understory of tree, shrub, herbaceous plant, bryophyte, and forest litter layers. Specimens of bryophytes were collected for identification in the laboratory.

The stand basal area was determined using the plot measurement method. The trees of each species were counted in each study plot and the stem diameter at breast height (dbh), i.e., 1.3 m above ground, of all trees in the plot with a stem diameter ≥ 10 cm was measured using a Mantax Black tree calliper (Haglöf Inc., Långsele, Sweden) [59]. The diameters were then converted to stem cross-sectional areas and the results were summed and divided by the plot area to obtain the stand basal area. Individual stand basal areas (*BAi*, m^2^) were computed using the following formula:$$BA_{i}=\pi \cdot {\left(\frac{d}{2}\right)}^{2}\cdot 10^{-4}$$
where *d* is the tree dbh in cm.

Based on the summed *BA_i_* measurements, the stand basal area (*ba*, m^2^/ha) of the study plots was calculated.

The stand density index (*SDI*) was computed using the following formula:$$SDI=tpha\cdot {\left(\frac{qmd}{25.4}\right)}^{1.605}$$
where *tpha* is the number of trees per hectare and *qmd* is the quadratic mean diameter in cm. 

The quadratic mean diameter (*qmd*, cm) was computed using the following formula:$$qmd=\sqrt{\frac{ba/tpha}{0.00007854}}$$
where *ba* is the stand basal area (m^2^/ha), *tpha* is the number of trees per hectare, and 0.00007854 is a constant.

The data on forest stand origin (natural vs. cultural), age, canopy height, and size of forest compartments were obtained from forest taxation documents, in cooperation with the State Forest Office. The measured tree diameters and standardised tree volume tables were used to estimate the volume of standing trees [60]. Photographs of different habitats in *S. globosum* localities are given in Figure 3.

Natural disturbances, including windblown debris, windbreak effects, snowbreak phenomena, and wild boar (*Sus scrofa*) digging as well as insect damage to trees, were investigated at all study plots. Additionally, anthropogenic disturbances such as forest management activities (e.g., salvage logging and thinning), forest paths, gravel and paved roads, and electric power transmission and telephone lines were also examined. The stumps and the trees damaged by insects, wind, or snow were counted and their diameters were measured for subsequent tree volume estimations. To evaluate the disturbance, the volume of trees blown down by wind or snow, damaged by insects, or felled was estimated as a percentage of the total stand volume within each plot. The tree volume table was used to estimate the volume of felled and fallen trees [60]. The area of disturbed forest floor by wild boar, forest management, or anthropogenic activities as a percentage of the total forest floor area was evaluated [61,62]. Photographs of observed disturbances in *S. globosum* localities are given in Appendix A.

The data from the governmental databases—namely, Cadastre of Protected Areas of the Republic of Lithuania (https://stvk.lt/map, accessed on 2 September 2023), Biodiversity database of State Service of Protected Areas under the Ministry of Environment of the Lithuanian Republic (https://biomon.lt/maps/index.php, accessed on 25 June 2023), and The Information System of Protected Species of State Service of Protected Areas under the Ministry of Environment of the Lithuanian Republic (https://sris.am.lt, accessed on 25 June 2023)—and a system of protected area management categories developed by the International Union for Conservation of Nature (IUCN) [63] were used to ascertain the protection status of *S. globosum* localities.

### 2.3. Data Analysis

All statistical analyses were performed using PAST v4.03 [64]. A one-way analysis of variance (ANOVA) was performed to examine the differences in soil chemical parameters among the habitats. Tukey’s honest significant difference (HSD) test was employed as a post hoc test, applied only if the ANOVA detected statistically significant differences between the habitats. Prior to the analysis, data were transformed either using arcsine or natural log transformations to meet the normality assumption. The assumption of normality was verified using the Shapiro–Wilk test. Additionally, the homoscedasticity assumption was assessed using the Levene test. A principal component analysis (PCA) based on a correlation matrix was used to determine the relationship between the environmental factors. The cover percentages were subjected to transformation using the arcsine function before being utilised in the PCA. A Venn analysis was performed to identify the number of disturbances among the study plots in *S. globosum* localities. 

## 3. Results

### 3.1. Distribution and Habitats of Sarcosoma globosum

The analysis of geospatial data from 28 *S. globosum* localities indicated a restricted distribution of this species in the north-eastern and eastern parts of Lithuania, between 54.8 and 55.8° N and between 24.5 and 26.3° E (Figure 2; Appendix A). The three largest localities were Labanoras 1 (ca. 11.1 ha), Antazavė 5, and Ilgašilis (both ca. 9.1 ha). The six largest localities accounted for 57.4% of the total area (ca. 78.9 ha) of *S. globosum* in the study area. The sizes given for localities are only estimates based on the areas of forest compartments with the presence of *S. globosum* and are not a precise measurement of the area occupied by the fungus under investigation.

All localities were situated in coniferous forests, both of natural (67.4% of all study plots) and cultural (32.6%) origin (Appendix A). The presence of Norway spruce (*Picea abies*) proved to be a positive indicator for the presence of *S. globosum* in the studied forest stands. *P. abies* was present in the canopy of all study plots (comprising 10–70% of the canopy layer cover) and in the understory of the majority of study plots (34 plots; 79.1%, comprising 10–50% of the understory layer cover). However, *P. abies* in both canopy and understory layers dominated only in 28 (65.1%) study plots. Another coniferous tree, *Pinus sylvestris*, was commonly observed in the study plots and this tree species dominated in the canopy of 13 (30.2%) study plots. Various sporadic deciduous tree species such as *Acer platanoides*, *Alnus glutinosa*, *Betula pendula*, *Populus tremula*, *Quercus robur*, *Sorbus aucuparia*, and *Tilia cordata* were also present in *S. globosum* localities.

Three EUNIS habitat types were identified at *S. globosum* localities. The most common habitat type was boreo-nemoral bilberry western spruce taiga (T3F14) (21 plots; 48.8% of all study plots), followed by continental tall-herb western spruce taiga (T3F44) (11 plots; 25.6%) and native fir, spruce, larch, and cedar plantations (T3N1) (10 plots; 23.3%) (Figure 3; Appendix A). The diversity of habitat types within a single locality was different; i.e., in five localities (namely, Antazavė 1, Ažvinčiai 1, Ilgašilis, Labanoras 1, and Minčia 2), two different habitat types were identified, while other localities were represented by single habitat type.

An analysis of the forest stand age structure revealed a rather large proportion of young Norway spruce stands of cultural origin—25.6% of the study plots was assigned to three age classes from 21 to 50 years (Figure 4; Appendix A). The youngest forest stand in which *S. globosum* was recorded was a 27-year-old Norway spruce plantation (plot Vt; observation in 2014). The majority of *S. globosum* growth sites were situated in older forests, with 16.3% in forests from 51 to 70 years old and 58.1% in forests from 71 to 151 years old. It was notable that in large S. *globosum* localities, the age of tree stands varied significantly; e.g., in the Labanoras 1 locality, it varied from 40 to 75 years; in the Ilgašilis forest locality, it varied from 32 to 145 years; and in the Antazavė 1 locality, it varied from 65 to 125 years.

The results of the forest tree measurements from the study plots and the subsequent calculations showed that the stand density (SDI) varied from 231.48 to 1573.88 (mean 713.43), the stand basal area (BA) varied from 11.41 to 85.52 m^2^/ha (mean 36.05), and the stand tree volume varied from 7.97 to 85.52 m^3^ (mean 22.76) (Appendix A). The largest SDI (1573.88), BA (85.52 m^2^/ha), and stand tree volume (85.52 m^3^) were in the 83-year-old Žiliškės forest locality. 

In total, 15 bryophyte species were identified on the forest floor in the study plots (Appendix A). The most common and dominant bryophyte species were *Hylocomium splendens* (occurring in 41 study plots and comprising 5–40% of the bryophyte layer cover in the study plots) and *Pleurozium schreberi* (respectively, 3 plots and 1–40%).

The PCA analysis using the correlation matrix did not show a clear relationship between habitat types and the environmental variables (Figure 5). Specifically, the differences in environmental variables among the *S. globosum* studied plots from three different habitat types did not reveal clear distinctions or groupings. The primary distinctions between old-growth and young forests in *S. globosum* growth sites lay in the understory composition, the cover of the herbaceous plant layer, and the density of forest litter. Additionally, these differences extended to whether the forest consisted of pure Norway spruce or included Scots pine.

The results from the soil analysis demonstrated that the forest stands in *S. globosum* localities predominantly grew on sandy loam soil and sand, specifically eutri-haplic arenosols and dystri-haplic arenosols, accounting for 51.2% and 30.2% of all localities, respectively (Appendix A). The study results also revealed that *S. globosum* grew on both soil with a rather thick organic layer (more than 9 cm in O horizon thickness) and rich organic carbon in the topsoil (more than 3%) and on soil with a rather thin organic layer (less than 7 cm in O horizon thickness) and poor organic carbon in the topsoil (less than 2%). The total nitrogen content in the topsoil was low (0.1–0.3%; mean 0.18%). The analysis of the main soil parameters showed that neither the O horizon thickness nor the concentrations of organic carbon and total nitrogen, nor the C:N ratio were significantly different between the studied habitat types (Figure 6). However, an analysis of the soil pH showed the differences between the boreo-nemoral bilberry western spruce taiga and continental tall-herb western spruce taiga habitats (*p* = 0.02). The soil of forests that included Scots pine was extremely acidic (the pH varied from 3 to 4.2), whereas the soil in the pure Norway spruce forests varied from extremely acidic to moderately acidic (the pH varied from 3.8 to 6).

### 3.2. Threats and State of Conservation

Four main groups of disturbances were identified in the studied plots of *S. globosum*. These were tree windthrow, windbreak, or snowbreak; tree damage by insects, usually the European spruce bark beetle (*Ips typographus*); forest logging during forest management practices, such as thinning or salvage logging; and disturbance of the forest floor by wild boar (*Sus scrofa*) digging or by logging machines (Appendix A; Figure 7 and Appendix A). The most frequent forest stand damage, which was observed in 90.4% of all study plots, was a result of windthrow, windbreak, or/and snowbreak. The most wind-affected localities were in the Ažvinčiai ancient woodland (5.6–36.5%). 

Both insect damage to trees and disturbed forest floors were observed in 64.3% of all study plots. In 26 study plots, the volume of trees damaged by insects as a percentage of the total stand volume varied from 0.2% to 9.4%. The most insect-affected stands were in Antazavė pinewood, where 35.7% of the trees on this site had been damaged by bark beetles. Considerably significant forest floor disturbance (20–45% of the plot area) was from wild boar in 5 study plots; a fully intact forest floor was observed in only 15 study plots. 

Traces of forest management activities were found in 75% of the study plots. In 13 study plots, the volume of trees felled as a percentage of the total plot stand volume was over 15%. The most damaged locality of *S. globosum* was in Paliūniškis forest, where 78.4% of trees had been felled during exhaustive salvage logging conducted at the beginning of 2022.

Almost all *S. globosum* localities were in protected areas; however, 8 study plots (18.6%) did not belong to any type of protected area (Appendix A). The majority of plots (33 plots; 76.7%) were in territories of national (IUCN protected area category II) and regional (category V) parks. Half of the plots (22; 51.2%) belonged to landscape, hydrographical, botanical–zoological, and botanical reserves (category IV). Only 5 study plots (11.6%) were in strictly protected areas of strict nature reserves (category Ia). Of the study plots, 35 (81.4%) belonged to the Natura 2000 network of protected areas under both the European Union Birds and Habitats Directives. Half of them (17 plots) were situated in two Natura 2000 habitat types—namely, 9010 Western taiga and 9050 Fennoscandian herb-rich forests with *Picea abies* (Figure 3; Appendix A). 

## 4. Discussion

The ascomycete species *S. globosum* was first described and illustrated in 1793 by the German naturalist Casimir Christoph Schmidel as *Burcardia globosa* Schmidel from fungus material collected in a pine forest near Erlangen in Germany [65]. This species has a wide distribution in the Northern Hemisphere, specifically in North America, Europe, and Asia [26]. In North America, it is a rare species and occurs only in the eastern parts of Canada and the USA on the ground in needle litter in both coniferous and mixed deciduous/coniferous forests with balsam fir (*Abies balsamea*)*,* jack pine (*Pinus banksiana*)*,* white pine (*P. strobus*), white spruce (*Picea glauca*), black spruce (*P. mariana*), white birch (*Betula papyrifera*), red maple (*Acer rubrum*), and trembling aspen (*Populus tremuloides*) [66,67]. The fungus is mainly restricted to old-growth forests and was listed among the species indicative of ecological integrity in humid boreal balsam fir pristine forest stands in the eastern part of Canada [25]. A recent report on the discovery of *S. globosum* in British Columbia, the westernmost province of Canada [68], requires a more detailed investigation. In Asia, the range of *S. globosum* lies mainly in northern part of Asia and includes taiga (boreal forest) zones in the West, Central, and western East Siberia of Russia [21,69,70]. The species occurs in swamped and well-drained sites; it inhabits forests dominated by coniferous trees such as Siberian pine (*Pinus sibirica*), Siberian spruce (*Picea ovata*), and Siberian fir (*Abies sibirica*) or it grows in European aspen (*Populus tremula*)-dominated forests with a coniferous undergrowth [21]. The species is characterised by a narrow ecological adaptation, fragmented distribution, and vulnerability to changes in environmental conditions; consequently, as an endangered species deserving protection, *S. globosum* was included in the regional and national Red Data Books of Russia [71,72]. A largely isolated subregion is situated within the Armenian Highlands of the Caucasus region in West Asia, specifically in Armenia [73].

In Europe, the *S. globosum* geographical distribution range roughly coincides with the limit of the last ice age in Europe and is associated with coniferous forests dominated by or with the presence of Norway spruce (*Picea abies*) in boreal, semi-boreal, and taiga forest zones and, by preference, in undisturbed pristine forests [22,26,32,35]. Fraiture and Otto [22] pointed out that the distribution of species appears to comprise two areas. The first is located in the northern and eastern parts of Europe (Sweden, Norway, Finland, Estonia, Latvia, Lithuania, the Kaliningrad region of Russia, the northernmost part of Poland, Belarus, Ukraine (the Kiev region), and the European part of Russia [16,20,23,24,26,27,28,30,31,32,33,34,35,36,37,38,39,40,41,42,71,74,75,76] (Artsdatabanken (https://artsdatabanken.no, accessed on 24 March 2024); SLU, the Swedish University of Agricultural Sciences (https://www.slu.se/en, accessed on 24 March 2024); and eElurikkus (https://elurikkus.ee/en, accessed on 24 March 2024)). The second area centres on Germany, Switzerland, Austria, Czechia, and Slovakia [20,22,29,77] (MykologischeDatenbank (https://www.pilzdaten-austria.eu, accessed on 24 March 2024) and SwissFungi (https://swissfungi.wsl.ch/en/, accessed on 24 March 2024)). The species is absent above the Arctic circle (northward of 67° N) [37]. The decrease in its frequency and extinction occurred in the 20th century in Central European countries [22] and, as mentioned in the Introduction, the fungus is believed to be extinct in Austria, Switzerland, Czechia, Slovakia, Germany, Poland, and the Kaliningrad region of Russia.

Due to the indispensable assistance of protected area personnel, foresters, naturalists, and other citizens, a total of 28 *S. globosum* localities were discovered between 2007 and 2021 in Lithuania. This result contrasts with the previous belief that the species was on the verge of extinction in the country [41]. We fully agree with the statement that mycology could benefit from increasing public interest in fungi [15,37,46]. Our present findings show that the localities of *S. globosum* in Lithuania are situated on the south-western edge of a shrinking geographical distribution range of this species in Europe. All *S. globosum* populations are concentrated in the eastern and north-eastern parts of Lithuania. It is worth mentioning that *P. abies*-dominated coniferous forests also grow in other parts of Lithuania, particularly in the western part of the country; however, *S. globosum* has not yet been recorded in these parts. Compared with the western part of Lithuania, which has the strong climatic influence of the Baltic Sea, the eastern and north-eastern parts of the country are characterised by colder winters with less precipitation (snow or rain) and warmer summers with more precipitation due to an east–west gradient, implying a greater distinction between seasons. Thus, the regions where *S. globosum* is currently present are colder than the rest of the country and with greater precipitation in autumn [78]. The Lithuanian, Swedish, and Belarusian *S. globosum* sites are in transition zones between two biogeographical regions; i.e., between the boreal and continental regions [51,52,79].

The available data on the soil chemical properties of *S. globosum* habitats show that *S. globosum* grows both in acidic and calcareous soil that is well supplied with water and rich in moss cover [16,22,24,35,74,79]. We observed that *S. globosum* in Lithuania grew on soil with a rather thick organic layer (from 5.9 to 12.9 cm in O horizon), with both rich and poor organic carbon in the topsoil (1.6–4.9%), a low nitrogen content in the topsoil (0.1–0.3%), extremely acidic to moderately acidic soil (the pH varied from 3 to 6), and on intact forest floors more or less covered with bryophytes, predominantly *Hylocomium splendens* and *Pleurozium schreberi.*

The diversity of habitat types in *S. globosum* sites in Europe is limited; all of them are characterised by forests with Norway spruce [16,22,24]. According to the Natura 2000 network habitat classification, *S. globosum* occurs in four habitat types—namely, 9010 Western taiga, 9410 Acidophilic *Picea* forests of montane to alpine levels, 9050 Fennoscandian herb-rich forests with *Picea abies*, and 9070 Fennoscandian wooded pastures [32,33]. Our study results confirmed these data. All *S. globosum* habitats in Lithuania were restricted to coniferous forests with the presence of *P. abies*. This fungus was found in both pure Norway spruce stands and mixed coniferous tree stands. The fungus in Lithuania occurred in two Natura 2000 network habitat types—namely, 9010 and 9050. We also assessed *S. globosum* habitats according EUNIS classifications and identified three habitat types. The most common habitat type was boreo-nemoral bilberry western spruce taiga (T3F14), followed by continental tall-herb western spruce taiga (T3F44) and native fir, spruce, larch, and cedar plantations (T3N1).

Forest age and continuity also matter for *S. globosum*. Research conducted in Finland and Sweden showed that *S. globosum* prefers forests between 60 and 100 years old [22,23,24,35,37,79]. The average age of stands was 70–80 years in the Kirov region of Russia [74] and the mean age of conifers in balsam fir forests was 87 years in the eastern part of Canada [25]. The fungus is considered to be an indicator species of forest ecological integrity and is generally associated with primeval and old-growth spruce or fir–spruce forests little affected by human activity [23,24,25]. Very rarely was *S. globosum* found in younger stands in Sweden [79]. Although our study broadly confirmed these observations, the analysis of forest stand age structures revealed a relatively high proportion of young Norway spruce stands with an anthropogenic origin (25.6% of study plots fell within age classes ranging from 21 to 50 years) within the Lithuanian localities of *S. globosum*. The youngest tree stand at a fungus site was a 27-year-old planted Norway spruce stand. The monoculture even-aged plantations of Norway spruce created since the end of the 20th century in Lithuania have proved to be suitable habitats for *S. globosum*.

It seems that changes in the vitality of Norway spruce in the region could be one of the most important threats to *S. globosum*. Due to its preferences for cool and moist climatic conditions, Norway spruce may become severely affected under global warming conditions. The projections based on future climate data present a dramatic retreat of the Norway spruce’s range. Current climate changes in Europe have led to changes in Norway spruce stands due to increased fire frequency, the proliferation of insect pests and pathogenic fungi, desiccation, wind disturbances, and browsing damage in young stands, which alter the structure of the vegetation cover and the distribution of spruce-associated organisms [80,81,82]. Past, ongoing, and future habitat declines in area and quality are estimated to negatively impact on all populations of *S. globosum* in Europe [26]. Moreover, forest management activities such as clear-cutting practices in old-growth forests could be the main threat to *S. globosum* populations [26]. Our observations showed that *S. globosum* survived in forest stands in which moderate selective logging (not intensive sanitary logging of dead or damaged Norway spruce trees) or forest thinning in young Norway spruce plantations was performed. 

*S. globosum* is a rare and endangered fungus in Europe; it is evaluated in the IUCN Global Fungal Red List as near-threatened (NT) and is nationally red-listed in 11 European countries and Armenia [26]. *S. globosum* has been protected in Lithuania since 1992 and was recently evaluated according the IUCN criteria as an endangered species (EN) [36,42]. The majority of localities are in protected areas of different categories; however, only two *S. globosum* localities are in strict nature reserves (IUCN protected area category Ia). The legal protection of the species includes other administrative measures, particularly the upload of species data to the Information System of Protected Species (ISPS) of the State Service of Protected Areas under the Ministry of Environment of the Lithuanian Republic (https://sris.am.lt/, accessed on 25 June 2023) as well as the development of action plans for threatened species and a restriction setting on management activities in species localities. Nevertheless, at the moment, not all species conservation measures are implemented; e.g., an action plan for the conservation of *S. globosum* has not yet been developed and the data at ISPS concerning known *S. globosum* localities are incomplete.

Another potential threat could be the picking of mushrooms [14] because, historically, *S. globosum* was used in Lithuania as a remedy [38,39,40] and it is still collected for medical purposes in some parts of Russia [21,74]. However, during the study period, we did not notice any reports concerning the picking of *S. globosum* fruiting bodies in Lithuania. From a biodiversity conservation perspective, *S. globosum* needs special attention to evaluate its conservation status and possible threats in future. All measures of in situ and ex situ conservation should be taken into consideration [15,83].

## Figures and Tables

**Figure 1 jof-10-00263-f001:**
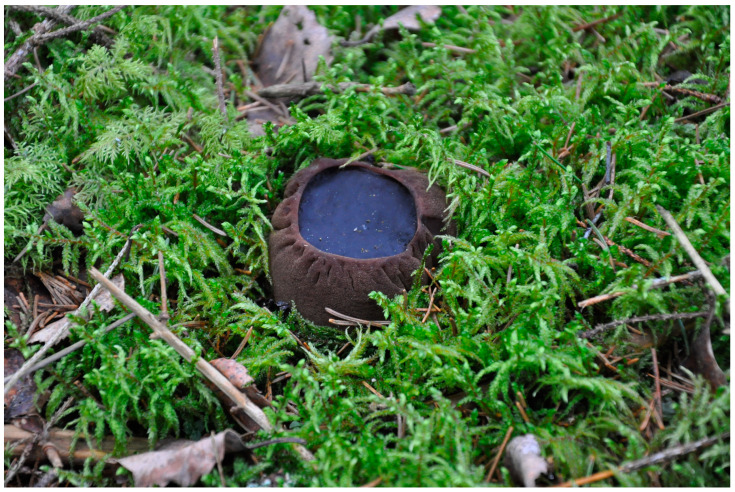
Apothecium of *Sarcosoma globosum* on needle litter between *Pleurozium schreberi* moss in a *Picea abies* stand (Kiauneliškis strict nature reserve, Lithuania; photo by E. Vabuolė, 2021).

**Figure 2 jof-10-00263-f002:**
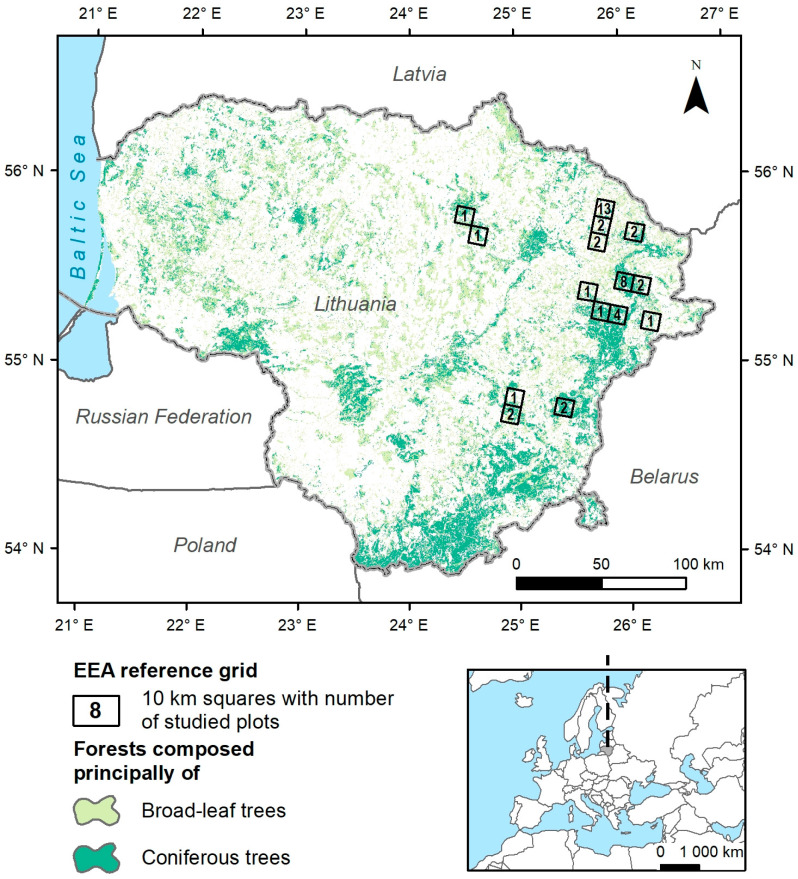
Gridded distribution of *Sarcosoma globosum* in Lithuania based on the European Environment Agency (EEA) and a reference grid resolution of 10 km.

**Figure 3 jof-10-00263-f003:**
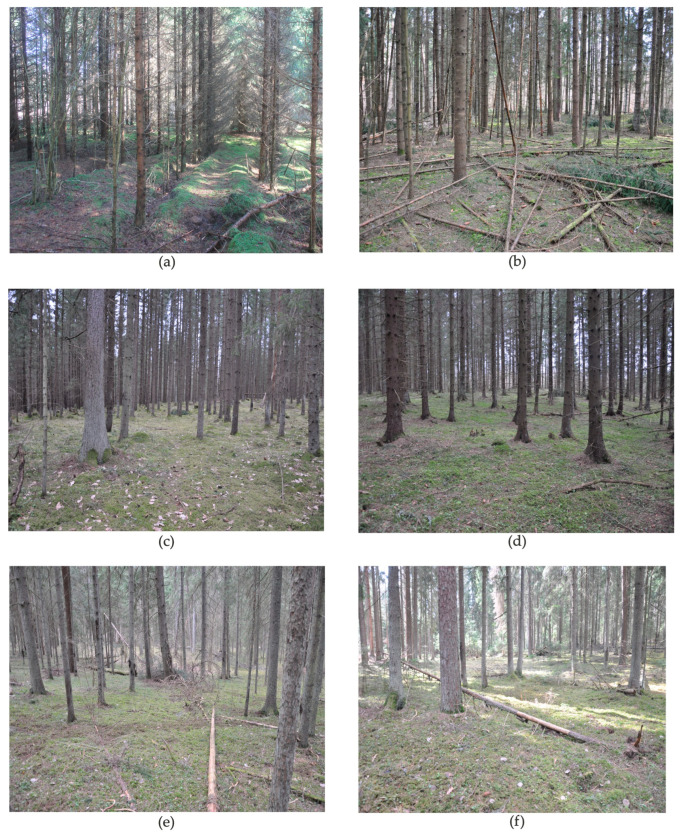
Habitat diversity in *Sarcosoma globosum* localities in Lithuania: (**a**) native fir, spruce, larch, and cedar plantations (T3N1), 35-year-old tree stand in Vyteniškės forest, plot Vt; (**b**) native fir, spruce, larch, and cedar plantations (T3N1), 46-year-old tree stand in Grabijolai forest, plot Grb; (**c**) boreo-nemoral bilberry western spruce taiga (T3F14), 60-year-old tree stand in Jaskoniškės forest, plot Ja1; (**d**) continental tall-herb western spruce taiga (T3F44), 60-year-old tree stand in Puščia forest, plot Pu; (**e**) continental tall-herb western spruce taiga (T3F44), 9050 Fennoscandian herb-rich forests with *Picea abies*, 125-year-old tree stand in Antazavė pinewood, plot An1d; (**f**) boreo-nemoral bilberry western spruce taiga (T3F14), 9010 Western taiga, 96-year-old tree stand in Ažvinčiai ancient woodland, plot Az2a (photos by E. Vabuolė, 2021).

**Figure 4 jof-10-00263-f004:**
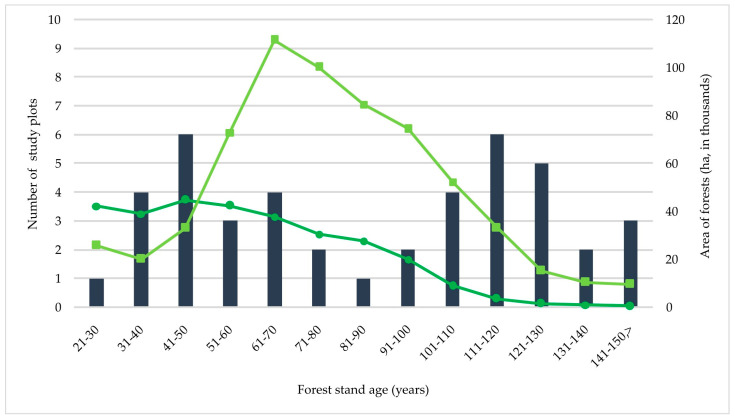
The number of *Sarcosoma globosum* study plots per forest stand age class (data from Appendix A; data on age and area of *Picea abies* (dark-green curve) and *Pinus sylvestris* (light-green curve) forests are from the State Forest Service database (https://amvmt.lrv.lt/lt/atviri-duomenys-1/misku-statistikos-leidiniai/valstybine-misku-apskaita/2022-01-01/, accessed on 10 July 2023)).

**Figure 5 jof-10-00263-f005:**
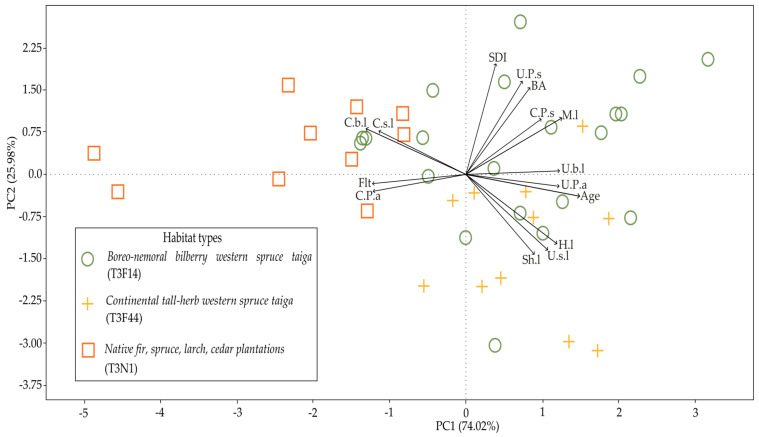
Principal component analysis (PCA) to determine the relationship between the habitat types and the environmental variables (forest stand age (years, age); stand density index (SDI); stand basal area (m^2^/ha, BA); canopy cover (%) of *Picea abies* (C.P.a), *Pinus sylvestris* (C.P.s), small-leaved trees (C.s.l), and broad-leaf trees (C.b.l); understory cover (%) of *P. abies* (U.P.a), *P. sylvestris* (U.P.s), small-leaved trees (U.s.l), and broad-leaf trees (U.b.l); shrubs (Sh.l), herbs (H.l), moss (M.l), and forest litter (Flt) layer cover (%) recorded at *Sarcosoma globosum* study plots.

**Figure 6 jof-10-00263-f006:**
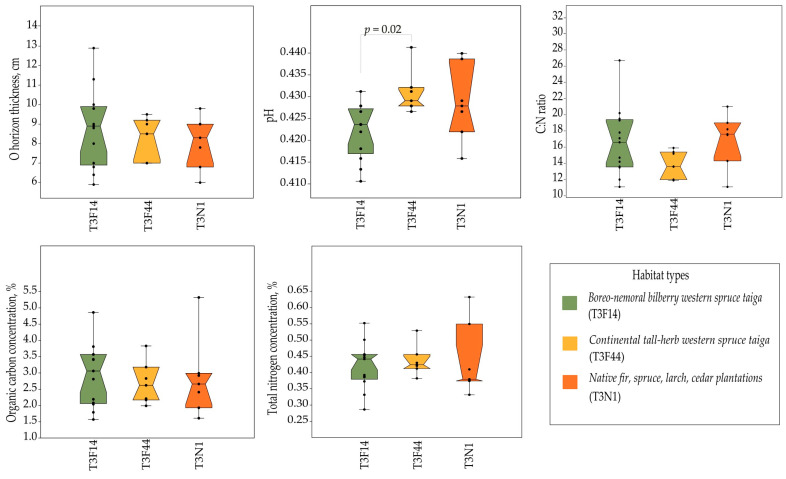
Comparison of soil parameters across three habitat types in the *Sarcosoma globosum* localities studied (*n* = 28) in Lithuania. One-way ANOVA and Tukey’s post hoc test assessed mean equality within each habitat, with significant differences denoted by Tukey’s *p*-values.

**Figure 7 jof-10-00263-f007:**
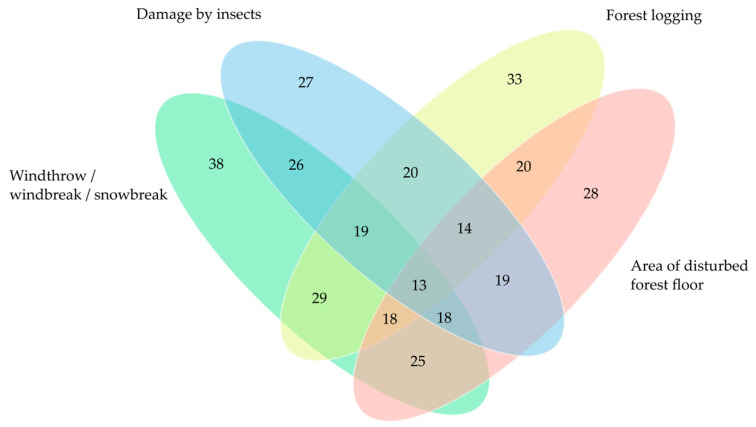
Venn diagram of main disturbances observed in *Sarcosoma globosum* study plots (*n* = 43) in Lithuania. The numbers in ellipses indicate the number of study plots where disturbances were observed.

## Data Availability

Data are contained within the article and Appendix A.

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
