# Peer review of "Habitat Diversity, Environmental Conditions, and Distribution of Endangered Fungus Sarcosoma globosum (Ascomycota) in Lithuania"

_jof, 2024, doi:10.3390/jof10040263_

Round 1

Reviewer 1 Report

The paper presents very well the physical characteristics of the habitats of Sarcosoma globosum in Lithuania. The meteorological data are, however, lacking, except some averages. The matter in question is, however, much about the influence of the climate change on the annual and seasonal occurrence of the species, and that is why those parameters should become dealed.

Data on the temperatures and precipitation specially in springtime, and even the amount and melting point of snow could tell more about the occurrence of the fungus in different years. At least here in North Finland disappearing of snow totally can be a threat, but such situation can be more serious in Lithuania.

I haven´t tried to correct small errors, and only appearance of Fennoscandia catches my eye.

Reviewer 2 Report

Thank you for inviting me to review this manuscript. The authors have undertaken a commendable job in describing and analysing the habitat diversity and environmental conditions of S. globosum in Lithuania. The manuscript aims to compare the characteristics of various habitats, investigate the distribution pattern of the fungus within Lithuanian forest ecosystems, and assess the impacts of both natural and anthropogenic disturbances on the localities of S. globosum. I am happy with the content of this paper and strongly advocate for its publication online. However, I have some suggestions that could enhance the clarity and rigor of the manuscript.

It is crucial for the readers to understand how the authors have accurately identified S. globosum, given the many species within the same genus. Could you elucidate the scientific methods employed for this identification? If molecular data were used, please detail the techniques and analyses. Alternatively, if the identification was based on morphological characteristics, please include a comprehensive description of these features in the materials and methods section.

 The statement in lines 141-142 regarding the sampling of topsoil from a single study plot in all S. globosum localities is ambiguous. It is unclear whether samples were collected from only one location or from one plot within each locality. Please provide a clearer description of your sampling strategy. Furthermore, if sampling was indeed limited to a single location, the discussions and conclusions drawn in lines 308-322 and 383-390 should be explicitly contextualized to reflect this scope.

In the discussion section, I recommend a comparison between the global distribution of S. globosum and your findings in Lithuania. Please review and incorporate existing literature on this fungus to situate your research within a global context. An exploration of whether there is a specific season that favors the fruiting of S. globosum would also enrich the discussion.
